# Gradient-Space Evolution Strategies for Efficient Diffusion Alignment

## Abstract

Reinforcement Learning (RL) has become essential for aligning diffusion models, yet existing methods often struggle with inefficiency and training instability. In this paper, we introduce **Gradient-Space Evolution Strategies (GS-ES)**, a novel framework that bridges Evolution Strategies and RL by reimagining alignment as a stepwise evolutionary process. Theoretically, we establish a duality between structured action-space sampling and gradient-space evolution, allowing us to estimate stable update directions directly from diffusion noise. This formulation effectively reduces the high variance inherent in standard policy gradients while eliminating the computational burden of maintaining reference models. Empirically, GS-ES demonstrates superior training efficiency and alignment quality across Text-to-Image (Flux) and Text-to-Video (Wan2.1) tasks, significantly outperforming standard GRPO-based baselines.

## 1. Introduction

Recent years have witnessed the transformative success of diffusion models (Ho et al., 2020; Lipman et al., 2023) in visual content creation (Labs et al., 2025; Wan et al., 2025). Mirroring the paradigm shift in Large Language Models (LLMs) (Brown et al., 2020), Reinforcement Learning from Human Feedback (RLHF) (Ouyang et al., 2022) has emerged as a crucial technique for aligning these generative models with human preferences. Theoretical advancements (Song et al., 2020) have unified the sampling processes of both Flow Matching and Diffusion Models within the framework of Stochastic Differential Equations (SDEs). Building upon this unification, recent works (Liu et al., 2025a; Xue et al., 2025) formulate the reverse SDE sampling trajectory as a Markov Decision Process (MDP),

successfully adapting algorithms like GRPO to diffusion alignment.

However, the prevailing Diffusion-RL methods (Liu et al., 2025a; Xue et al., 2025; He et al., 2025; Wang et al., 2025; Li et al., 2025) remain largely **LLM-centric**, heavily borrowing from discrete token generation frameworks without fully exploiting the intrinsic properties of diffusion models. These methods treat the denoising trajectory as a sequence of discrete actions and rely heavily on a Reference Model to enforce constraints. This design choice imposes a significant computational and memory burden, slowing down the iterative training process and limiting both scalability and the performance ceiling.

We argue that even within the MDP formulation, the sampling process of diffusion models possesses unique characteristics distinct from autoregressive LLMs. By designing alignment algorithms that respect these properties, we can unlock both superior training efficiency and alignment performance. Specifically, we identify two key properties of diffusion models that have been underutilized in RL contexts:

- **Step-wise Differentiability:** Unlike the discrete nature of text tokens, each step of the diffusion denoising process is fully differentiable. This implies that exploration in the latent action space can be analytically mapped to the parameter gradient space via the model's Jacobian (see Sec. 4.3).

- **Fixed-Horizon Iterative Refinement:** Diffusion sampling follows a fixed number of function evaluations (NFE) with a predetermined trajectory length. This structured, fixed-horizon nature distinguishes it from the variable-length rollouts in LLMs, allowing for more flexible and effective algorithmic designs—such as the integration of population-based evolutionary strategies.

Building on these insights, we propose **Gradient-Space Evolution Strategies (GS-ES)**, a novel alignment framework that reimagines Diffusion RL through the lens of Evolution Strategies (Rechenberg, 1978; Wierstra et al., 2014; Salimans et al., 2017). GS-ES directly leverages the diffusion model's differentiability to perform efficient exploration. Specifically, within a single gradient-enabled forward pass per population sample, we perform multiple local

[1]Anonymous Institution, Anonymous City, Anonymous Region, Anonymous Country. Correspondence to: Anonymous Author <anon.email@domain.com>.

Preliminary work. Under review by the International Conference on Machine Learning (ICML). Do not distribute.

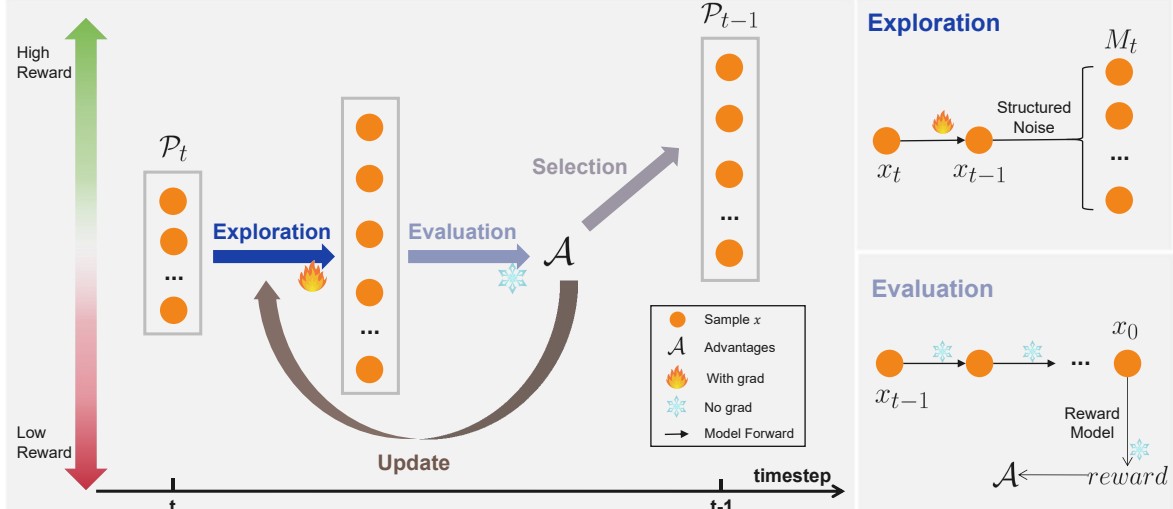

*Figure 1.* **Framework of Gradient-Space Evolution Strategies (GS-ES).** The diagram illustrates the evolutionary iteration from timestep $t$ to $t - 1$. At each step, we sequentially perform **Exploration**, **Evaluation**, **Selection**, and **Update**.

perturbations (Action Noise) in the latent space. Through backpropagation, these low-dimensional perturbations in the action space effectively serve as "probes" that guide high-dimensional evolution in the gradient space. Furthermore, inspired by classic ES, we incorporate **Population Selection** (dynamic trajectory pruning) and **Structured Exploration** into the sampling process. This allows our framework to discard the Reference Model and KL penalties entirely, operating as a memory-efficient, on-policy update algorithm that procedurally mirrors ES but mathematically retains the rigorous gradient estimation of RL.

Our contributions are summarized as follows:

- **A Novel Diffusion-Native Alignment Paradigm:** We propose GS-ES, the first framework to seamlessly integrate Evolution Strategies with Diffusion RL. By shifting from an "LLM-adaptation" mindset to a "Diffusion-native" design, our work offers a fresh perspective that reimagines diffusion alignment as an unconstrained evolutionary process, eliminating the need for Reference Models and KL divergence calculations.

- **Action-Gradient Duality for Efficient Alignment:** We establish a formal connection between structured noise in the action space and evolutionary search in the gradient space. This derivation, rooted in standard score function identities, demonstrates that our action-space perturbation strategy effectively serves as a variance-reduced estimator of the policy gradient, providing a theoretical basis for the algorithm's superior stability compared to standard methods.

- **Comprehensive Empirical Validation:** Through ex-

tensive experiments and detailed ablation studies, we validate the necessity and efficacy of each algorithmic component. Our results demonstrate that GS-ES achieves state-of-the-art alignment performance with faster convergence across both Text-to-Image and Text-to-Video tasks, proving that GS-ES is both viable and highly scalable.

## 2. Related Work

### 2.1. Alignment for Diffusion Models

A substantial body of research has emerged focused on aligning diffusion and flow matching models with human preferences. Training-free alignment methods (Song et al., 2023; Tang et al., 2024; Yeh et al., 2025) inject preference signals during inference without retraining. Although flexible, these methods often incur high latency and offer limited control. Offline approaches (Wallace et al., 2024; Yang et al., 2024; Liang et al., 2024; Zhang et al., 2025; Liu et al., 2025b) adapt Direct Preference Optimization (DPO) to visual generation, utilizing paired data to optimize a classification-like objective. Meanwhile, online methods (Black et al., 2023; Fan et al., 2023) leverage Proximal Policy Optimization (PPO) to optimize against reward models directly. Representing the current state-of-the-art, GRPO-based approaches such as Flow-GRPO (Liu et al., 2025a) and DanceGRPO (Xue et al., 2025) integrate Group Relative Policy Optimization into flow matching frameworks. By converting deterministic ODEs to stochastic SDEs, they enable divergent sampling essential for RL exploration.

However, these methods largely force-fit standard RL paradigms onto diffusion models, underutilizing their intrinsic properties. In contrast, we reimagine alignment from an evolutionary perspective, proposing a framework natively designed to exploit these unique characteristics for efficient alignment.

## 2.2. Evolution Strategies in Deep Learning

Evolution Strategies (ES) (Huning, 1976) encompass a class of black-box optimization algorithms inspired by natural evolution. Unlike gradient-based methods, ES optimizes an objective by maintaining a population of parameter vectors, evolving them through mutation (perturbation) and fitness-based selection. A pivotal variant, **Natural Evolution Strategies (NES)** (Wierstra et al., 2014; Salimans et al., 2017), estimates update directions by applying stochastic perturbations directly to the model parameters. This creates a fundamental distinction in policy optimization: while traditional Reinforcement Learning (RL) relies on exploration in the **Action Space**, ES drives exploration directly in the **Parameter Space**. To further enhance sample efficiency, subsequent works (Plappert et al., 2017; Choromanski et al., 2018) have introduced structured exploration techniques (e.g., orthogonal sampling) within the ES framework.

However, directly applying parameter-space ES to modern diffusion models is intractable due to the billion-scale dimensionality of the parameters. Our GS-ES framework bridges this gap by transposing the evolutionary process into the **Gradient Space**. By leveraging the model's differentiability, we integrate core ES principles—specifically population selection and structured noise—to guide efficient alignment.

## 3. Preliminaries

### 3.1. Reinforcement Learning for Diffusion Models

**Diffusion as an MDP.** The sampling processes of both Diffusion Models and Flow Matching can be unified through the perspective of the reverse-time Stochastic Differential Equation (SDE) (Song et al., 2020). Recent works (Fan et al., 2023; Black et al., 2023; Liu et al., 2025a; Xue et al., 2025) discretize this generation trajectory into a Markov Decision Process (MDP), where the latent state $x_t$ evolves to $x_{t-1}$ via a policy $\pi_\theta$. Crucially, these methods rely on the standard Gaussian assumption to define the single-step transition probability as:

$$\pi_\theta(x_{t-1}|x_t) = \mathcal{N}(x_{t-1}; \mu_\theta(x_t, t), \sigma_t^2 \mathbf{I}), \quad (1)$$

where $\mu_\theta$ represents the model-predicted mean (derived from velocity $v_\theta$ or noise $\epsilon_\theta$) and $\sigma_t$ denotes the noise schedule.

**Policy Gradient & GRPO.** The objective of diffusion RL is to maximize the expected reward $J(\theta) = \mathbb{E}_{x_0 \sim \pi_\theta}[R(x_0)]$. Based on the standard Policy Gradient theorem (Williams, 1992), the gradient is computed as:

$$\nabla_\theta J_{\text{PG}}(\theta) = \mathbb{E}_{\tau \sim \pi_\theta} \left[ \sum_{t=1}^{T} \nabla_\theta \log \pi_\theta(x_{t-1}|x_t) \cdot A_t \right], \quad (2)$$

where $A_t$ represents the advantage (e.g., the return $R(x_0)$). To mitigate high variance and ensure stability, state-of-the-art methods (Liu et al., 2025a; Xue et al., 2025) adopt a **GRPO-style** mechanism. Specifically, they modify Eq. 2 by incorporating an importance sampling ratio or a KL divergence penalty to constrain policy updates within a trust region.

### 3.2. Evolution Strategies (ES)

**Natural Evolution Strategies (NES).** Unlike gradient-based RL which optimizes a stochastic policy, NES (Wierstra et al., 2014; Salimans et al., 2017) maximizes the expected objective value over a search distribution in the parameter space. The gradient of the Gaussian-smoothed objective $J(\theta) = \mathbb{E}_{\xi \sim \mathcal{N}(0, \mathbf{I})}[F(\theta + \sigma\xi)]$ is estimated as:

$$\nabla_\theta J_{\text{ES}}(\theta) \approx \frac{1}{\sigma N} \sum_{i=1}^{N} \xi_i \cdot F(\theta + \sigma\xi_i). \quad (3)$$

**The Action-Parameter Duality.** Comparing Eq. 2 and Eq. 3 highlights a fundamental structural symmetry. First, regarding **Exploration**, RL relies on stochasticity in the **Action Space** (sampling $\pi_\theta$), whereas ES relies on stochasticity in the **Parameter Space** (perturbing $\theta$ via $\xi$). Second, regarding **Evaluation**, the fitness function $F(\cdot)$ in ES plays a functionally identical role to the Advantage function $A_t$ in RL—both serving as scalar signals to reweight the update direction based on the quality of the perturbation.

## 4. Method

This section presents **Gradient-Space Evolution Strategies (GS-ES)**. We first provide a high-level overview of the algorithmic framework in Sec. 4.1. Subsequently, Sec. 4.2 details the implementation of key components, including exploration, evaluation, selection, and update. Finally, Sec. 4.3 provides a theoretical analysis proving the duality between our action-space sampling and gradient-space evolution.

### 4.1. Overview of the GS-ES Framework

We propose **GS-ES**, a reference-free, efficient alignment framework tailored for diffusion models. The core procedure is outlined in Algorithm 1. Structurally, our framework mirrors the classic Evolution Strategies loop while being natively integrated into the differentiable diffusion sampling

**Algorithm 1** Gradient-Space Evolution Strategies (GS-ES)

---

1: **Input:** Diffusion Model $\pi_\theta$, Reward Model $R$
2: **Hyperparameters:** Budget $M_{\text{fix}}$, Group Size $G$
3: Initialize population $\mathcal{P}_T = \{x_T\}$, where $x_T \sim \mathcal{N}(0, \mathbf{I})$
4: **for** $t = T, \dots, 1$ **do**
5:     **Gradient-Enabled Forward:**
        Compute $\mu_\theta(x, t)$ for all $x \in \mathcal{P}_t$
6:     **Exploration:**
        Expand $\mathcal{P}_t$ into candidates $\{x_{t-1}^{(j)}\}_{j=1}^b$
        (where each parent spawns $G$ children)
7:     **Evaluation (Gradient-free):**
        Get Reward: $r^{(j)} = R(x_{t-1}^{(j)})$
        Compute Advantage $A^{(j)}$
8:     **Update Model:**
        Backpropagate fitness-weighted gradients from
        all $M_t$ candidates
9:     **Select:**
        Calculate $P_{t-1}$ using $M_{\text{fix}}$ and $G$ (Eq. 4, 5)
        Select survivors: $\mathcal{P}_{t-1} \leftarrow \text{Select}(\{x_{t-1}^{(j)}\}, P_{t-1})$
10: **end for**

---

step. As illustrated in Figure 1, unlike traditional Diffusion RL approaches that optimize the denoising trajectory at the **trajectory level**, GS-ES shifts the paradigm to **point-level optimization**. We reimagine the alignment process as a **Step-wise Evolutionary Optimization** process, introducing evolutionary selection and gradient space sampling mechanisms at each sampling step.

The core workflow of GS-ES iterates through three phases at each denoising timestep $t$:

1. **Exploration:** Instead of sampling a single next state, we generate a population of candidate actions $\{x_{t-1}^{(i)}\}_{i=1}^{M_t}$ by applying structured perturbations to the current mean prediction.

2. **Evaluation:** We estimate the potential reward (fitness) for each candidate branch.

3. **Selection & Update:** We apply evolutionary selection operators to filter the fittest candidates (forming a population of size $P_{t-1}$) as the basis for the next timestep. Simultaneously, we aggregate the fitness-weighted gradients from all $M_t$ candidates to update the model parameters. Crucially, as derived in Sec. 4.3, this action-space sampling effectively functions as structured sampling in the parameter gradient space.

Notably, this framework eliminates the need for an external Reference Model, a separate sampling policy, or importance sampling ratios, significantly streamlining the alignment process compared to prior GRPO-based methods.

## 4.2. Key Algorithmic Components

This section details the four key components of GS-ES: exploration, evaluation, selection, and update. While our framework supports a broad spectrum of implementations for these operators (akin to modular designs in RL and ES), we focus here on established strategies and diffusion-specific optimizations to validate the core methodology.

### 4.2.1. EXPLORATION: STRUCTURED NOISE SAMPLING

Recall that the single-step transition is modeled as a Gaussian (Eq. 1). We generate the candidate population $\{x_{t-1}^{(i)}\}_{i=1}^{M_t}$ using **Structured Noise Sampling** strategies adapted from ES literature (Salimans et al., 2017; Choromanski et al., 2018; Plappert et al., 2017). Specifically, we investigate random noise, antithetic sampling, and orthogonal sampling. These structured perturbations serve as efficient local probes, maximizing directional diversity in the action space with minimal redundancy.

### 4.2.2. EVALUATION: MULTI-STEP ODE

To obtain valid fitness signals for the intermediate states $x_{t-1}^{(i)}$, we follow standard diffusion sampling protocols. For each candidate, we utilize a gradient-free ODE solver to advance the generation process by the remaining steps to produce a converged image $\hat{x}_0^{(i)}$, which is then scored by the Reward Model. For the fitness metric, we explore various formulations, including Rank-based normalization and Group-Relative normalization. Empirically, we find that the **Group-Relative** formulation (Shao et al., 2024) provides robust signals for guiding evolution by normalizing rewards within each local branching group.

### 4.2.3. SELECTION MECHANISMS

The evolutionary process at each timestep is governed by three components: the **Sampling Budget** ($M_t$), the **Population Size** ($P_t$), and the **Selection Operator**. Unlike standard GRPO which retains all samples, GS-ES introduces a selection pressure to prune low-quality trajectories, balancing exploration width with computational constraints.

**Dynamic Sampling Budget** ($M_t$). Our goal is to maintain a consistent computational cost (wall-clock time) across all timesteps while matching the total budget of a standard GRPO baseline with a fixed sample size $M_{\text{fix}}$. Given a fixed generation horizon $T$, the evaluation cost at step $t$—which integrates the ODE from $t - 1$ to $0$—is proportional to the lookahead length $L_t = t - 1$. Using a fixed sample size leads to an imbalanced load (expensive at $t = T$, cheap at $t = 1$). For convenience, we use $M_{\text{fix}}$ as the primary hyperparameter to control the algorithmic computational budget. Our experiments (see Sec. 5.3) demonstrate that this compute-balanced design significantly outperforms a

fixed branch size given the same computational cost.

Specifically, we first calculate the total computational cost (NFE) for the baseline fixed schedule: $\mathcal{C}_{\text{total}} \approx \sum_{t=1}^{T} M_{\text{fix}} \cdot (t-1) \approx \frac{1}{2} M_{\text{fix}} T^2$. Distributing this total cost evenly across $T$ steps yields a constant per-step budget $\mathcal{B} = \frac{1}{2} M_{\text{fix}} T$. The dynamic total sampling budget $M_t$ at timestep $t$ is thus derived to satisfy $M_t \cdot (t-1) \approx \mathcal{B}$:

$$M_t = \text{round}\left(\frac{M_{\text{fix}} \cdot T}{2 \cdot \max(1, t-1)}\right). \qquad (4)$$

This schedule allocates fewer samples when evaluation is expensive and significantly more samples when evaluation is cheap ($t \to 1$).

**Population Sizing ($P_t$).** To enable group-relative advantage estimation, we organize the $M_t$ samples into groups to calculate reliable advantage baselines. We introduce the **Group Size** ($G$) as a fixed hyperparameter, representing the number of branches generated per parent state. Consequently, the number of distinct trajectories retained for the next step ($P_t$) is deterministically derived by dividing the total budget by the group size:

$$P_t = \max\left(1, \left\lfloor \frac{M_t}{G} \right\rfloor\right). \qquad (5)$$

This formulation naturally results in an evolutionary dynamics: starting with a narrow population (small $P_T$) for coarse global search, and expanding to a broad population (large $P_1$) for fine-grained local optimization.

**Candidate Selection.** Once $M_t$ candidates are generated and evaluated (organized into $P_t$ groups of size $G$), the selection operator determines the basis states for the subsequent timestep $t - 1$. Within each group, we compute the advantages and apply a selection strategy (e.g., **Softmax** (Sutton et al., 1998; Mahfoud & Goldberg, 1995), **Tournament Selection** (Goldberg & Deb, 1991), or **Greedy**) to select one survivor per group. This ensures that the population size remains consistent with the budget constraints while adaptively filtering out lower-reward branches.

### 4.2.4. UPDATE: GRADIENT AGGREGATION

Finally, we update the model parameters by aggregating the gradients from the explored population. Referencing the standard Policy Gradient objective (Eq. 2), we compute the weighted sum of the policy gradients for each candidate, using their calculated fitness (advantage) as weights:

$$\Delta\theta \propto \sum_{i=1}^{M_t} A^{(i)} \nabla_\theta \log \pi_\theta(x_{t-1}^{(i)}|x_t). \qquad (6)$$

As we formally prove in Sec. 4.3, although this update is computed via action-space samples $x_{t-1}^{(i)}$, it is mathematically equivalent to performing structured evolution in the

**gradient space**, thereby bridging the gap between RL exploration and ES stability.

### 4.3. Theoretical Analysis: Gradient-Space Evolution

In this section, we provide a theoretical justification for our method. We formally prove that although GS-ES operates by perturbing states in the action space, it functionally performs an evolutionary update directly in the gradient space.

**Proposition 4.1** (Action-Gradient Duality). *In a diffusion denoising step with a Gaussian policy $\pi_\theta(x_{t-1}|x_t) = \mathcal{N}(\mu_\theta, \sigma_t^2 \mathbf{I})$, the REINFORCE gradient estimate derived from an action-space perturbation $\epsilon$ is mathematically equivalent to a Natural Evolution Strategy (NES) update projected onto the parameter space via the model's Jacobian $\mathbf{J}_\mu = \nabla_\theta \mu_\theta$.*

*Proof Sketch.* (See Appendix A for the rigorous derivation). Recall the Policy Gradient objective (Eq. 2) where the update direction depends on the score function $\nabla_\theta \log \pi_\theta(x_{t-1}|x_t)$. Given the reparameterization $x_{t-1} = \mu_\theta(x_t) + \sigma_t \epsilon$, where $\epsilon \sim \mathcal{N}(0, \mathbf{I})$, we use the chain rule to derive:

$$\begin{aligned}\nabla_\theta \log \pi_\theta(x_{t-1}|x_t) &= \nabla_\theta \left(-\frac{\|x_{t-1} - \mu_\theta\|^2}{2\sigma_t^2}\right) \\ &= \frac{1}{\sigma_t}(\nabla_\theta \mu_\theta)^\top \epsilon.\end{aligned} \qquad (7)$$

Substituting this into the gradient estimator, the update rule becomes:

$$\Delta\theta \propto \mathbb{E}_\epsilon \left[A(\epsilon) \cdot (\mathbf{J}_\mu^\top \epsilon)\right]. \qquad (8)$$

Here, the term $\mathbf{g}_\epsilon = \mathbf{J}_\mu^\top \epsilon$ represents a specific direction vector in the parameter space, analytically mapped from the action noise via the model's Jacobian. In practice, this enables efficient optimization requiring only a single gradient-enabled forward pass and standard backpropagation per step, bypassing the need to materialize the computationally prohibitive Jacobian matrix.

**Evolution in Gradient Space.** This result offers a fresh perspective on diffusion alignment. Traditional ES blindly samples perturbations in the high-dimensional parameter space (Eq. 3). In contrast, GS-ES samples perturbations in the low-dimensional action space and leverages the diffusion model's differentiable structure ($\mathbf{J}_\mu$) to project them into informed update directions $\mathbf{g}_\epsilon$. This allows GS-ES to enjoy the stability of Evolution Strategies while leveraging the efficiency of first-order backpropagation, effectively performing evolution on the gradients themselves.

*Table 1.* Quantitative comparison on Text-to-Image (Flux) and Text-to-Video (Wan2.1) tasks. We evaluate different algorithms on specific reward models based on the same base model. We report the raw scores and the improvement over the base model. To analyze training efficiency, we also detail the computational cost per denoising step, reporting the count of **Gradient-Enabled / Gradient-Free** forward passes and the average wall-clock **Time** per step. **Bold** indicates the best result, and underline indicates the second best.

*Text-to-Image (Base Model: Flux-dev)*

| Method | Alignment Metrics | | | Efficiency (per step) | |
|---|---|---|---|---|---|
| | PickScore ↑ | HPSv2 ↑ | HPSv3 ↑ | Grad / Free | Time (s) |
| Base Model (Flux) | 22.58 | 29.55 | 13.91 | - | - |
| Flow-GRPO | 23.40 (+0.82) | 37.33 (+7.78) | 15.02 (+1.11) | 432 / 768 | 740 |
| DanceGRPO | 23.33 (+0.75) | 37.81 (+8.26) | 15.00 (+1.09) | 432 / 768 | 740 |
| **GS-ES (Ours)** | **24.27** (+**1.69**) | **39.75** (+**10.20**) | **15.56** (+**1.65**) | 26 / 1466 | 720 |

*Text-to-Video (Base Model: Wan2.1-T2V-1.3B)*

| Method | Alignment Metrics | | | Efficiency (per step) | |
|---|---|---|---|---|---|
| | PickScore ↑ | HPSv2 ↑ | VideoAlign ↑ | Grad / Free | Time (s) |
| Base Model (Wan2.1) | 18.92 | 16.45 | 0.72 | - | - |
| Flow-GRPO | 19.61 (+0.69) | 24.34 (+7.89) | 0.99 (+0.27) | 576 / 960 | 360 |
| DanceGRPO | 19.35 (+0.43) | 19.93 (+3.48) | 0.93 (+0.21) | 576 / 960 | 360 |
| **GS-ES (Ours)** | **20.31** (+**1.39**) | **28.79** (+**12.34**) | **1.46** (+**0.74**) | 37 / 2130 | 390 |

## 5. Experiments

### 5.1. Experimental Setup

To ensure a rigorous and fair comparison, our experimental protocols largely follow those established by Dance-GRPO (Xue et al., 2025), extensively validated by both the official authors and the broader research community. We evaluate GS-ES primarily on text-to-image (T2I) generation and extend it to text-to-video (T2V) to demonstrate scalability.

**Base Models & Baselines.** For T2I, we employ **Flux-dev** (Labs et al., 2025) as the base model. We benchmark against state-of-the-art RL methods, specifically **Flow-GRPO** (Liu et al., 2025a) and **DanceGRPO** (Xue et al., 2025). Both methods adapt Group Relative Policy Optimization (GRPO) to diffusion models but diverge in implementation details: Flow-GRPO incorporates a KL divergence penalty to constrain policy updates, whereas DanceGRPO omits the KL term in favor of a shared noise initialization strategy to enhance training stability. For T2V, we utilize **Wan2.1-T2V-1.3B** (Wan et al., 2025) to evaluate video alignment capabilities.

**Objectives & Datasets.** Consistent with the DanceGRPO setting, we align models using PickScore (Kirstain et al., 2023), HPSv2 (Wu et al., 2023), and HPSv3 (Ma et al., 2025) as reward models. For benchmarking, we use the PickScore metric on the Pick-a-Pic v1 dataset (Kirstain et al., 2023) and HPSv2/v3 metrics on the HPDv2 dataset (Wu et al., 2023). We evaluate the algorithmic performance using the corresponding test sets, consisting of 500 prompts for Pick-a-Pic and 400 prompts for HPDv2, respectively. For the T2V task, all data are curated from the VidProM (Wang

& Yang, 2024) dataset, where we select a dedicated subset of 400 prompts as the test set. For reward computation, we conduct separate experiments using three distinct reward models: PickScore, HPSv2, and VideoAlign (Liu et al., 2025a). For VideoAlign (Liu et al., 2025a), we use only the Text Alignment (TA) score, due to its superior stability compared to MQ and VQ metrics.

**Configuration.** Consistent with DanceGRPO, we set the sampling steps $T = 16$ for Text-to-Image tasks. To match the per-step wall-clock time, we employ a dynamic sampling budget with $M_{\text{fix}} = 16$ (Sec. 4.2.3). Additionally, we utilize **Orthogonal Noise Exploration** and **Softmax Selection**. For Text-to-Video generation, we increase the sampling steps to $T = 25$. All experiments are conducted on 8×H100 GPUs. Detailed hyperparameters and implementation details are provided in Appendix C.1.

### 5.2. Main Results

**Quantitative Performance.** Table 1 summarizes the alignment performance across both T2I and T2V tasks. GS-ES achieves superior results compared to all GRPO-based baselines across all metrics. In the T2I task, GS-ES demonstrates a substantial margin over the strongest baseline (Dance-GRPO or Flow-GRPO), achieving a **+1.69** gain in PickScore and a remarkable **+10.20** improvement in HPSv2 over the base model. This indicates that our step-wise evolutionary pressure effectively navigates the generation manifold towards higher human preference. Crucially, in the T2V domain, GS-ES significantly boosts the HPSv2 score from 16.45 to **28.79** (+12.34), largely outperforming DanceGRPO (+3.48). This result validates the scalability and stability of our framework in complex video generation tasks.

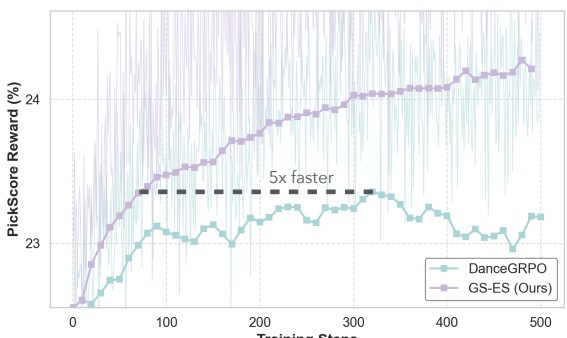

*Figure 2.* Training efficiency comparison on PickScore (Flux). Solid lines indicate test set performance, while semi-transparent lines represent training dynamics. GS-ES (Ours) demonstrates significantly faster convergence and higher final rewards.

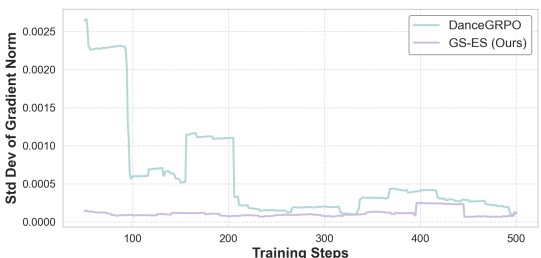

*Figure 3.* Sliding standard deviation of gradient norms (window size=50). GS-ES maintains lower volatility, indicating the superior stability of our gradient estimation.

**Training Efficiency.** As shown in Table 1, GS-ES maintains a per-step computational cost comparable to GRPO-based baselines. However, regarding convergence speed, Figure 2 illustrates that GS-ES demonstrates superior efficiency on the Flux PickScore task. Specifically, GS-ES reaches the peak performance level of DanceGRPO approximately **5.0×faster**. Moreover, upon full convergence, our method continues to improve, achieving a final reward gain that is **2.2×higher** than that of the baseline. We attribute this efficiency to the effective exploration in the gradient space, which provides the model with significantly more stable and informative gradient update directions.

**Gradient Stability.** To empirically verify the stability of our gradient estimation (discussed in Sec. 4.3), we visualize the sliding standard deviation of the gradient norms in Figure 3. As observed, GS-ES exhibits consistently lower volatility compared to DanceGRPO. Given that both methods maintain comparable gradient magnitudes, this lower deviation serves as a direct indicator of a more stable gradient estimator. We attribute this to our **Gradient-Space Evolution** mechanism: by aggregating directional information from the population, GS-ES mitigates the stochasticity inherent in standard policy gradients, serving as a robust proxy for the true gradient direction.

**Qualitative Comparison.** As shown in Figure 4, GS-ES generates images with higher visual fidelity and better prompt alignment compared to baselines. Additional visual samples are available in Appendix C.4.

### 5.3. Ablation Studies

We conduct extensive ablation studies to validate the effectiveness of each algorithmic component. Unless stated otherwise, all comparisons are performed relative to our default configuration on Flux (GS-ES with Dynamic Sampling Budget $M_{\text{fix}} = 16$, Softmax Selection, and Orthogonal Noise).

**Benefit of Compute-Balanced Population.** We validate the efficacy of our **Constant Compute Budget** scheduler (Sec. 4.2.3). As shown in Table 2, we compare the performance impact of different population schedules relative to our default GS-ES setting. Under the same computational budget (1.00× step time), the **Fixed** $M$ strategy leads to a significant performance drop (-0.51 PickScore), proving that our dynamic schedule is more effective than a uniform allocation. Furthermore, increasing the budget parameter to $M_{\text{fix}} = 24$ yields only marginal gains (+0.03) at a considerably higher cost (1.28×), while reducing it to $M_{\text{fix}} = 8$ degrades performance, indicating that $M_{\text{fix}} = 16$ is a sweet spot for efficiency.

*Table 2.* Ablation on Population Schedule (vs. Default $M_{\text{fix}} = 16$).

| Schedule (Setting) | Δ PickScore | Δ HPSv3 | Time |
|---|---|---|---|
| Fixed $M = 16$ | -0.51 | -0.33 | 1.00× |
| Varying $M_{\text{fix}} = 8$ | -0.42 | -0.24 | 0.67× |
| Varying $M_{\text{fix}} = 24$ | +0.03 | +0.08 | 1.28× |

*Table 3.* Ablation on Selection (vs. Softmax).

| Selection Strategy | Δ PickScore | Δ HPSv3 |
|---|---|---|
| Greedy | -0.25 | -0.23 |
| Random Selection | -0.06 | -0.15 |
| Tournament Selection | -0.17 | -0.21 |

**Effectiveness of Selection Mechanisms.** We analyze the impact of different selection operators (Sec. 4.2.3) in Table 3. Compared to Softmax, **Greedy** selection, which retains only the highest-reward sample, leads to a significant performance drop (-0.25 PickScore), likely due to the severe loss of diversity. Conversely, **Random** selection fails to provide sufficient evolutionary pressure to guide alignment. Notably, while **Tournament** selection is often effective in traditional ES, it underperforms in our diffusion context.

**Impact of Structured Exploration.** Finally, we investigate the noise injection strategy used during the exploration phase (Sec. 4.2.1), where we replace standard Gaussian noise in the SDE transition with structured alternatives. Ta-

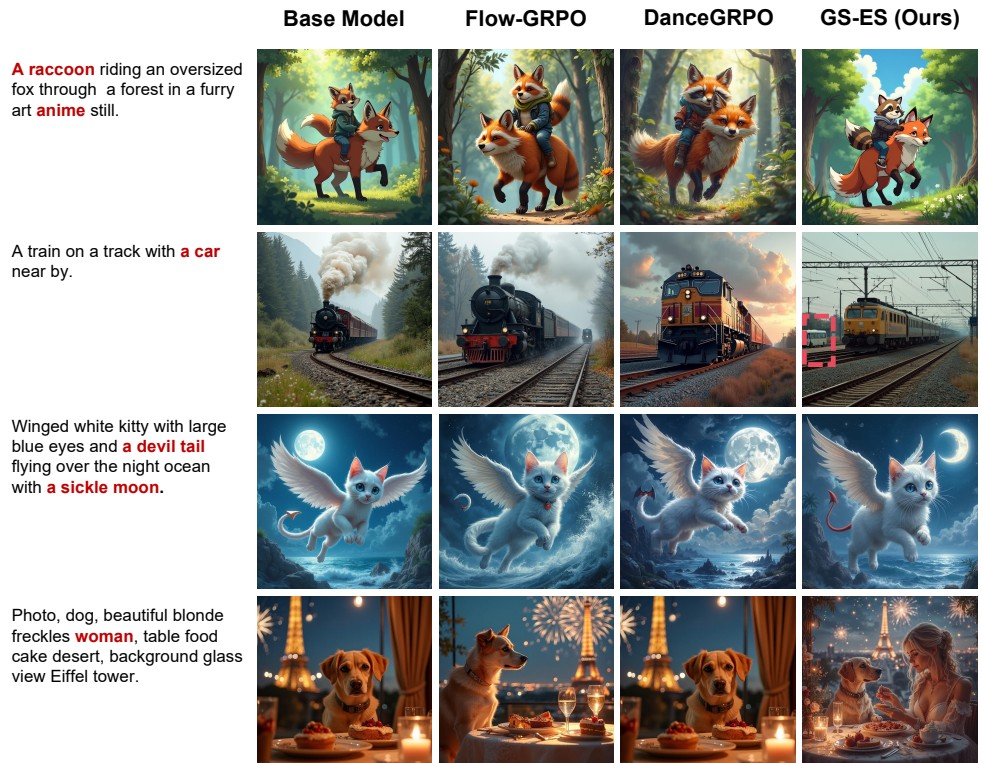

|  | Base Model | Flow-GRPO | DanceGRPO | GS-ES (Ours) |

**A raccoon** riding an oversized fox through a forest in a furry art **anime** still.

A train on a track with **a car** near by.

Winged white kitty with large blue eyes and **a devil tail** flying over the night ocean with **a sickle moon**.

Photo, dog, beautiful blonde freckles **woman**, table food cake desert, background glass view Eiffel tower.

*Figure 4.* Qualitative results. **Top two rows**: HPSv3 optimization; **Bottom two**: PickScore optimization.

ble 4 compares these strategies against our default Orthogonal noise. The results show that structured noise (both Orthogonal and Antithetic) consistently outperforms standard Random Gaussian noise. This empirical evidence indicates that structured action-space sampling is uniquely suitable for our GS-ES framework. Notably, structured noise yielded negligible gains for GRPO baselines, as their algorithmic design renders them insensitive to noise geometry.

*Table 4.* Ablation on Exploration Noise (vs. Orthogonal).

| Noise Type | $\Delta$ PickScore | $\Delta$ HPSv3 |
|---|---|---|
| Antithetic Sampling | -0.08 | -0.19 |
| Random Gaussian | -0.15 | -0.36 |

### 5.4. Limitation and Discussion

**Hyperparameter Sensitivity.** As an evolutionary framework, GS-ES introduces specific hyperparameters (e.g., $M_{\text{fix}}$ and $G$). While our experiments demonstrate that the compute-balanced setting yields stable results across various tasks, the tuning space is intrinsically larger than standard GRPO baselines. However, we argue that these parameters are inherent to the flexibility of ES, offering finer control over the exploration-exploitation trade-off. Future work

could explore auto-tuning mechanisms to mitigate this burden.

**Extended Analysis and Visuals.** Due to space constraints, we provide comprehensive supplementary materials in the Appendix. Specifically, Appendix B offers a detailed discussion with concurrent works (He et al., 2025; Li et al., 2025). Appendix C presents comprehensive supplementary experimental content.

## 6. Conclusion

In this paper, we introduced **Gradient-Space Evolution Strategies (GS-ES)**, a novel framework that bridges the gap between Evolution Strategies and Reinforcement Learning for diffusion alignment. Theoretically, we established the duality between action-space structured sampling and gradient-space evolution, legitimizing our efficient update rule in diffusion alignment. Empirically, GS-ES demonstrates superior alignment performance and training efficiency on both Text-to-Image and Text-to-Video tasks. We hope this work inspires further research into evolutionary perspectives on generative model alignment, particularly in leveraging the differentiable nature of diffusion processes.

## Impact Statement

This paper presents work whose goal is to advance the field of Machine Learning. By introducing a significantly more efficient alignment framework, our work has the potential to democratize access to high-quality generative models by lowering computational barriers. However, we acknowledge that advancements in Text-to-Image and Text-to-Video generation carry inherent risks regarding potential misuse, such as the creation of misleading synthetic content. We believe these risks are common to the field of generative AI and are not specifically exacerbated by our algorithmic contributions. We do not feel that there are other specific societal consequences that must be highlighted here.

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

## A. Theoretical Derivation

In this appendix, we provide the complete proof for **Proposition 4.1**, establishing the connection between Action-Space Noise in Diffusion RL and Parameter-Space Evolution Strategies.

### A.1. Setup: Diffusion as a Gaussian Policy

As established in Sec. 3.1, we consider the discretized reverse-time SDE as a Markov Decision Process. At timestep $t$, the diffusion model predicts a mean $\mu_\theta(x_t, t)$. The policy $\pi_\theta(x_{t-1}|x_t)$ is defined as a Gaussian distribution:

$$\pi_\theta(x_{t-1}|x_t) = \frac{1}{(2\pi\sigma_t^2)^{d/2}} \exp\left(-\frac{\|x_{t-1} - \mu_\theta(x_t, t)\|^2}{2\sigma_t^2}\right), \tag{9}$$

where $d$ is the dimension of the latent space (Action Space). In our GS-ES framework, we sample candidate actions using the reparameterization trick:

$$x_{t-1} = \mu_\theta(x_t, t) + \sigma_t\epsilon, \quad \text{where } \epsilon \sim \mathcal{N}(0, \mathbf{I}). \tag{10}$$

### A.2. Derivation of the Score Function

We seek to compute the score function $\nabla_\theta \log \pi_\theta(x_{t-1}|x_t)$, which is the core component of the REINFORCE algorithm. Taking the logarithm of the policy:

$$\log \pi_\theta(x_{t-1}|x_t) = -\frac{1}{2\sigma_t^2}\|x_{t-1} - \mu_\theta(x_t, t)\|^2 + C, \tag{11}$$

where $C$ is a constant independent of $\theta$.

To find the gradient with respect to parameters $\theta$, we apply the chain rule. Let $\mathbf{J}_\mu = \nabla_\theta \mu_\theta \in \mathbb{R}^{d \times |\theta|}$ be the Jacobian matrix of the model output with respect to its parameters. Recall that for a vector function $\mu(\theta)$, the gradient of a scalar composition $L(\mu(\theta))$ is given by $\nabla_\theta L = \mathbf{J}_\mu^\top \nabla_\mu L$.

First, compute the gradient with respect to $\mu$:

$$\nabla_\mu \log \pi_\theta = \nabla_\mu \left(-\frac{\|x_{t-1} - \mu\|^2}{2\sigma_t^2}\right) = \frac{x_{t-1} - \mu}{\sigma_t^2}. \tag{12}$$

Substituting the reparameterization $x_{t-1} - \mu = \sigma_t\epsilon$:

$$\nabla_\mu \log \pi_\theta = \frac{\sigma_t\epsilon}{\sigma_t^2} = \frac{\epsilon}{\sigma_t}. \tag{13}$$

Now, mapping this back to the parameter space via the Jacobian:

$$\nabla_\theta \log \pi_\theta(x_{t-1}|x_t) = \mathbf{J}_\mu^\top (\nabla_\mu \log \pi_\theta) = \frac{1}{\sigma_t}\mathbf{J}_\mu^\top \epsilon. \tag{14}$$

### A.3. Equivalence to Projected Evolution Strategies

The standard update rule for our GS-ES algorithm (based on Policy Gradient) is:

$$\Delta\theta_{GS-ES} \propto \mathbb{E}_\epsilon \left[A(\epsilon)\nabla_\theta \log \pi_\theta(x_{t-1}|x_t)\right]. \tag{15}$$

Substituting Eq. 14 into this update rule:

$$\Delta\theta_{GS-ES} \propto \frac{1}{\sigma_t}\mathbb{E}_\epsilon \left[A(\epsilon)(\mathbf{J}_\mu^\top \epsilon)\right]. \tag{16}$$

Now, compare this to a hypothetical zero-order Natural Evolution Strategy (NES) that would operate directly in the parameter space. If we were to perturb the parameters directly as $\theta' = \theta + \nu\xi$ (where $\xi$ is parameter noise), the NES update would be $\Delta\theta_{NES} \propto \mathbb{E}_\xi[F(\theta')\xi]$.

Eq. 16 reveals that our method is equivalent to an ES update where the parameter perturbation is not random $\xi$, but a specific direction $\xi_{proj} = \mathbf{J}_\mu^\top \epsilon$. This $\mathbf{J}_\mu^\top \epsilon$ is precisely the **projection of the action-space noise onto the parameter manifold**.

### A.4. Implication for Structured Exploration

This derivation proves why structured noise in the action space translates to efficient optimization. Consider **Antithetic Sampling** where we sample pairs $\{\epsilon, -\epsilon\}$. The aggregate gradient update is proportional to:

$$\Delta\theta \propto A(\epsilon)\mathbf{J}_\mu^\top \epsilon + A(-\epsilon)\mathbf{J}_\mu^\top (-\epsilon) = (A(\epsilon) - A(-\epsilon))\mathbf{J}_\mu^\top \epsilon. \tag{17}$$

Because the mapping $\epsilon \to \mathbf{J}_\mu^\top \epsilon$ is linear, the structured properties (e.g., zero-mean, orthogonality) enforced in the low-dimensional action space ($d \approx 10^4$) are analytically preserved when projected into the high-dimensional parameter space ($|\theta| \approx 10^9$). This allows GS-ES to achieve the stability of Antithetic ES without the computational intractability of perturbing billion-scale parameters directly. □

## B. Extended Discussion

- **Minimalist REINFORCE vs. Constrained GRPO.** Although recent efficient frameworks (He et al., 2025; Li et al., 2025) also leverage multi-step ODE solvers to accelerate reward estimation, most concurrent methods still fundamentally operate on a GRPO-based objective. This necessitates a Reference Model to compute KL divergence or importance sampling ratios to ensure training stability. In contrast, GS-ES effectively strips away these constraints, optimizing a pure REINFORCE objective without auxiliary models. We achieve stability not via external constraints, but through **Evolutionary Selection**—an implicit trust region formed by actively filtering low-reward trajectories. This design results in a significantly lighter memory footprint and a simpler implementation.

- **Step-wise Evolution vs. Trajectory Optimization.** While methods like TempFlow (He et al., 2025) sample multiple points, they typically wait until the end of a full generation to perform standard gradient updates. In contrast, GS-ES actively filters out low-quality paths at each individual denoising step. By retaining only the high-reward candidates to continue to the next step, our method ensures that the model wastes no computation on poor trajectories. Consequently, both the model optimization and subsequent exploration are consistently constrained to the most promising regions of the generation manifold.

- **Exploiting Noise Geometry.** A unique advantage of our framework is the efficacy of structured noise. In GRPO-based methods, the impact of noise design is often masked by the clipping mechanism or importance ratios. In GS-ES, due to our direct REINFORCE formulation (Action-Gradient Duality), structured noise (e.g., Antithetic or Orthogonal) directly translates to variance-reduced gradient estimation, providing a tangible performance boost that is largely absent in other frameworks.

- **Inspiring Evolutionary Perspectives.** Beyond immediate performance gains, GS-ES serves as a conceptual bridge connecting the rich literature of Evolution Strategies with generative model alignment. By framing the denoising process as an evolutionary dynamic, we hope to inspire the community to rethink diffusion alignment through an ES lens. This perspective paves the way for integrating more advanced evolutionary mechanisms into future alignment frameworks.

## C. Extended Experimental Results

In this section, we provide comprehensive supplementary materials to facilitate a deeper understanding of our experimental results and ensure reproducibility. First, we detail the exact hyperparameters and baseline configurations in Appendix C.1. Next, we describe the precise implementation logic of our selection operators in Appendix C.2, followed by an additional ablation study on advantage estimation. Finally, we present extensive qualitative comparisons in Appendix C.4 to demonstrate the robustness of our method.

### C.1. Experimental Settings

To ensure reproducibility, **we provide the anonymous source code in the supplementary material and will make the complete repository publicly available**. All experiments were conducted on NVIDIA 8×H100 GPUs.

For fair comparison, the core design philosophy of our base hyperparameter configuration is to align the wall-clock time per step between GS-ES and the baseline DanceGRPO. Tables 5 and 6 detail the specific hyperparameters used for the DanceGRPO baseline and our GS-ES method, respectively.

**Baseline Configurations.**

- **DanceGRPO:** We follow the community-verified settings utilizing shared noise initialization to ensure stability.

- **Flow-GRPO:** To ensure a rigorous comparison, we adhere to the same hyperparameter configuration as DanceGRPO (e.g., learning rate, batch size). We set the KL divergence coefficient to $\beta = 0.04$.

**Ours: GS-ES.** We set the baseline budget parameter $M_{\text{fix}} = 16$ to approximate the total computational cost of DanceGRPO's generations. This determines the dynamic sampling budget $M_t$ for each step as described in Sec. 4.2.3.

For the Text-to-Video task, we configure the resolution to $512 \times 512 \times 21$.



*Table 5.* Hyperparameters for DanceGRPO.

| Hyperparameter | Value |
| --- | --- |
| *Optimization* | |
| Batch Size (per GPU) | 2 |
| Grad Accum. Steps | 24 |
| Learning Rate | 1e-5 |
| Weight Decay | 1e-4 |
| Max Grad Norm | 0.01 |
| Grad Checkpointing | True |
| Use EMA | True |
| *Diffusion & GRPO* | |
| Resolution | $720 \times 720$ |
| Timestep Fraction | 0.6 |
| Group Size | 24 |
| Adv Clip Max | 5.0 |
| Ratio Clip ($\epsilon$) | 1e-4 |
| Init Same Noise | True |
| Shift | 3.0 |

*Table 6.* Hyperparameters for GS-ES (Ours).

| Hyperparameter | Value |
| --- | --- |
| *Optimization* | |
| Batch Size (per GPU) | 1 |
| - | - |
| Learning Rate | 1e-5 |
| Weight Decay | 1e-4 |
| Max Grad Norm | 0.01 |
| Grad Checkpointing | True |
| Use EMA | True |
| *Evolutionary Strategy* | |
| Resolution | $720 \times 720$ |
| $M_{\text{fix}}$ | 16 |
| Selection Strategy | Softmax |
| Exploration Noise | Orthogonal |
| Advantage Type | Group |
| Group Num | 10 |
| Adv Clip Max | 5.0 |
| Shift | 3.0 |



## C.2. Implementation Details of Selection Operators

To facilitate reproduction, we provide the precise logic for the selection mechanisms discussed in Sec. 4.2.3. Let $\mathcal{C} = \{1, \ldots, N\}$ be the set of candidate indices and $R \in \mathbb{R}^N$ be their corresponding rewards. We aim to select a subset of indices $\mathcal{I} \subset \mathcal{C}$ with $|\mathcal{I}| = K$.

**Softmax Selection (Default).** We model the selection probability using a Boltzmann distribution. The probability of selecting the $i$-th candidate is given by:

$$P(i) = \frac{\exp(R_i/\tau)}{\sum_{j=1}^{N} \exp(R_j/\tau)} \tag{18}$$

where $\tau$ is the temperature parameter (default $\tau = 1.0$). Crucially, to ensure diversity in the surviving population, we perform sampling without replacement. This guarantees that the selected $K$ parents are unique, preventing the population from collapsing into identical trajectories early in the generation process.

**Tournament Selection.** We implement a standard tournament selection with a "without replacement" constraint to maintain diversity. The process for selecting $K$ parents is as follows:

1. Maintain a pool of available indices, initially $\mathcal{P} = \mathcal{C}$.

2. For $k = 1$ to $K$:
    - Randomly sample a subset (tournament) $\mathcal{T} \subset \mathcal{P}$ of size $N_{tourn} = 3$.
    - Identify the winner $i^* = \arg\max_{j \in \mathcal{T}} R_j$.
    - Add $i^*$ to the selected set $\mathcal{I}$.

- Remove $i^*$ from the pool $\mathcal{P}$ to ensure it is not selected again.

While Tournament selection is robust in traditional ES, our ablation (Table 3) suggests that the probabilistic nature of Softmax better aligns with the stochastic gradients in diffusion alignment.

## C.3. Additional Ablation: Rank-based Advantage

In addition to the selection strategy, we explored alternative formulations for the advantage estimation. Specifically, we tested replacing the Group Relative Advantage (normalizing rewards within a group) with Rank-based Advantage, a technique common in ES strategies. In this setting, rewards are converted to ranks (normalized to $[-0.5, 0.5]$) before gradient aggregation. However, empirical results on the Flux-PickScore task showed that Rank-based advantage degraded performance, resulting in a **-0.17** drop in PickScore compared to our default Group Advantage. We hypothesize that the magnitude of the reward difference (lost in ranking) contains valuable signal for fine-grained alignment in high-dimensional diffusion models.

## C.4. Additional Visual Results

In this section, we provide an extensive qualitative comparison to further demonstrate the robustness of GS-ES. Figure 5 showcases generated samples across a diverse set of prompts, comparing our method against the Base Model and baselines.

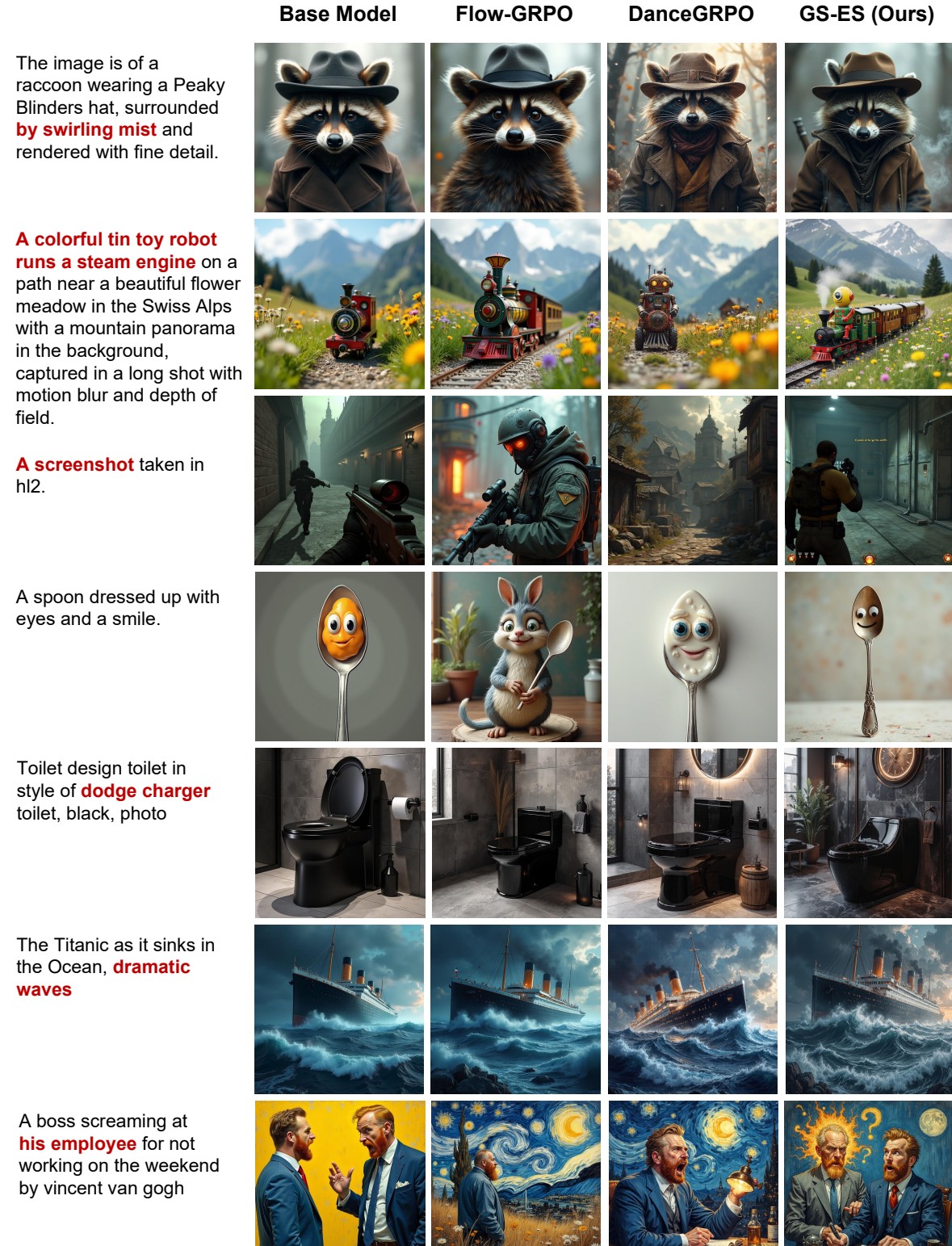

Figure 5. **Extended qualitative comparison of generated samples.** Columns correspond to the Base Model, Flow-GRPO, DanceGRPO, and GS-ES (Ours). The top four rows display results optimized for the HPSv2 reward, while the bottom three rows show results optimized for PickScore.

