# OpenReview forum: "Gradient-Space Evolution Strategies for Efficient Diffusion Alignment"
_ICML.cc/2026/Conference — Submitted to ICML 2026_

### Official Review · Reviewer_L4h3 · 2026-03-02

**Soundness:** 2
**Presentation:** 3
**Significance:** 2
**Originality:** 3
**Overall Recommendation:** 4
**Confidence:** 3

**Summary:**

This paper proposes an alignment paradigm that integrates Evolution Strategies (ES) with Diffusion Reinforcement Learning. It argues that diffusion model sampling possesses unique properties (step-wise differentiability and fixed-horizon iterative refinement) that are underutilized in autoregressive LLMs. Leveraging an action-parameter duality, the method reframes alignment as evolution in gradient space. Results are shown on text-to-image and text-to-video tasks, benchmarked against Flow-GRPO and DanceGRPO using reward models including PickScore, HPSv2/v3, and VideoAlign.

**Compliance With Llm Reviewing Policy:**

Affirmed.

**Final Justification:**

I completely understand the challenges posed by computational cost and the brevity of the rebuttal period. I will raise my score despite remaining concern on model architecture ablations.

**Key Questions For Authors:**

1. I would love to hear the authors' feedback on some suggestions above.

2. The evaluation phase rolls out each candidate sequentially, which could be quite expensive - would it be possible for candidates to be batched for the ODE rollout (if some conditions are shared)?

3. How much difference does the hybrid ES exploration + policy gradient update bring, compared with simply performing GRPO with structured noise and beam search?

4. The reward function implicitly depends on the model as candidates are evaluated by forward rollout of the current model, but the rewards are then treated as constants when computing advantages. This could create a bias - if the model shifted slightly, would the same candidates still be ranked in the same order?

**Limitations:**

Yes.

**Strengths And Weaknesses:**

Strength:

- The exploration mechanism is well-aligned with the ES literature. The use of structured noise is principled under the ES framework, and the paper has good ablation design for it.

- The Population Selection is well-motivated (in fact I think the paper could have a bit more discussion on this part) - could be by some observations that early denoising steps determine coarser structure (where exploration is most valuable) while late steps refine details (where exploitation suffices).

- Proposed method is straightforward and easy to implement.


Weakness:

- The current architecture processes denoising steps in order and accumulates gradients across several steps before the optimizer actually updates the weights. The normalizing factor treats all step-gradients as equally important, but the staleness of gradients from early steps could be a concern. The paper needs more justification on the choice.

- The model claims to eliminate the need of PPO-style ratio clipping or KL regularization by providing implicit regularization through "actively filtering low-reward trajectories", but this needs to be demonstrated with FID or similar metrics. Or above all, train on one reward model and evaluate on a different one to demonstrate the elimination of reward hacking.

- The results table does not mention standard deviations or confidence intervals.

- The qualitative comparison figures show prompts where competing methods miss specific keywords while this model captures them. While PickScore and HPSv2/v3 are good hybrid, comprehensive scores to report, it would also be important to backup the qualitative figures with compositional fidelity scores (e.g. CLIP, etc.)

---

> ### Author Rebuttal · Authors · 2026-03-31
>
> We thank Reviewer L4h3 for appreciating our ES-inspired exploration mechanism, ablation designs, and method simplicity. We address your concerns below.
>
> ### 1. Population Selection & Early-Stage Exploration (Strength 2)
>
> We strongly agree with your insight that early denoising steps determine coarser structures (where exploration is most valuable), while late steps refine details (where exploitation suffices).
>
> We believe that our population selection operation makes the GS-ES framework highly extensible, allowing heuristic modifications from other algorithms to be seamlessly integrated:
>
> - **Early-Stage Prioritization:** As discussed with Reviewer cVaG (Point 5), we implemented **GS-ES-Fast**, a variant that prioritizes the sampling budget exclusively on the early denoising window. This theoretically grounded allocation yields the best performance-to-compute trade-off.
>
> - **Sample Diversity:** Recent works highlight that sample diversity is crucial for the effectiveness of diffusion RL. Because of our explicit selection phase, transitioning from ranking candidates solely by reward to integrating sample diversity into the selection criteria is a straightforward and highly promising avenue for future work.
>
> ### 2. Gradient Staleness & Model Shift (Weakness 1 & KeyQuestion 4)
> You raised a critical point regarding gradient staleness and potential bias if the model shifts during sequential evaluation.
>
> First, in our implementation, the candidates at each step are generated and evaluated by the *latest* model via multi-step ODE rollout, ensuring the reward ranking perfectly reflects the current policy.
>
> Second, to prevent gradient staleness across steps, we strictly control the Gradient Accumulation Steps (GAS). By setting the GAS to cover the *entire* sampling process (a standard practice recommended by Flow-GRPO and DanceGRPO), the model parameters remain strictly frozen during the generation and evaluation of the whole trajectory. The gradients from all steps are averaged before a single optimization step is taken.
>
> Finally, as demonstrated in our KL-divergence analysis (**Exp 1** in response to Reviewer HSVu), our ES selection creates an implicit trust region that naturally prevents aggressive policy shifts, further mitigating staleness concerns.
>
> ### 3. Reward Hacking & Fidelity Evaluation (Weakness 2 & Weakness 4)
> We agree that cross-domain and rule-based metrics are essential to prove the elimination of reward hacking. Please see our responses to other reviewers for these exact experiments:
> - **Exp 2 (Cross-Reward):** In our **Response to Reviewer HSVu (Point 3)**, we trained GS-ES purely on PickScore and evaluated it on HPSv2, HPSv3, and CLIP scores. This demonstrates strong out-of-domain generalization.
> - **OOD Rule-Based Evaluation:** We evaluated GenEval when trained *purely on PickScore*. Our **Response to Reviewer cVaG (Point 6)** confirms this maintains rule-following compositionality without explicit KL constraints.
> - **In-Domain Rule-Based Evaluation (Exp 3):** Our **Response to Reviewer HSVu (Point 4)** provides results when trained directly on GenEval, showing stable improvements.
>
> Additionally, we computed CLIP scores for the qualitative results in Figure 4 (Left to Right: Base, Flow-GRPO, DanceGRPO, **GS-ES**):
> - Row 1: 39.98 | 41.78 | 41.87 | **44.07**
> - Row 2: **46.82** | 45.95 | 46.56 | 46.41
> - Row 3: 27.01 | 29.37 | 33.49 | **34.56**
> - Row 4: 40.87 | 40.26 | 41.11 | **47.26**
>
> CLIP variations are largely prompt-dependent. While challenging compositional prompts cause slight fluctuations across all methods (Row 2), GS-ES achieves significantly pronounced fidelity improvements elsewhere. We will add these scores to the revised figures.
>
> ### 4. Standard Deviations (Weakness 3)
> Given the massive computational cost of Diffusion RL, we (along with prior works like Flow-GRPO) initially omitted standard deviations. However, we acknowledge their importance for rigorous evaluation and will do our best to include confidence intervals in the main results (Table 1) of the final camera-ready version.
>
> ### 5. Batched ODE Rollout (KeyQuestion 2)
> Yes, candidates can absolutely be batched for the ODE rollout if they share conditions, subject only to VRAM constraints. In fact, any higher-order sampling method is compatible with our framework, provided the evaluation yields a sufficient reward signal for ranking.
>
> ### 6. Comparison with GRPO + Beam Search (KeyQuestion 3)
> Intuitively, GS-ES can indeed be viewed as a formal, ES-grounded restructuring of a "GRPO + structured noise + beam search" baseline, but with the critical addition of population selection and gradient-space mapping.
>
> To quantify this difference, we included this exact baseline in our empirical comparison against recent SOTA methods (BranchGRPO, TempFlowGRPO, DiffusionNFT). Please refer to **Exp 4 in our Response to Reviewer ADvX**. The results show that GS-ES achieves superior alignment performance.

---

> > ### Author Rebuttal · Reviewer_L4h3 · 2026-04-03
> >
> > I truly appreciate the authors' efforts in providing additional context and experiments. While the extra empirical comparisons (Exp 4) show only marginal differences, it is difficult to fully assess their significance without reporting the variance. This makes it harder to fully support the claim of the "critical addition of population selection and gradient-space mapping".
> >
> > That said, I completely understand the challenges posed by computational cost and the brevity of the rebuttal period. I will raise my score.

---

### Official Review · Reviewer_cVaG · 2026-03-12

**Soundness:** 2
**Presentation:** 3
**Significance:** 3
**Originality:** 2
**Overall Recommendation:** 4
**Confidence:** 4

**Summary:**

1. This paper argues that even within the MDP formulation, the sampling process of diffusion models possesses unique characteristics distinct from autoregressive LLMs.
2. This paper proposes GS-ES, a Novel Diffusion-Native Alignment Paradigm.
3. This paper establishes a formal connection between structured noise in the action space and evolutionary search in the gradient space.

**Compliance With Llm Reviewing Policy:**

Affirmed.

**Final Justification:**

All my concerns are resolved by the authors' rebuttal, and I will raise my score.

**Key Questions For Authors:**

Question:
1. While the authors claim to introduce a "new paradigm" in the introduction, the framework appears to be more of a sophisticated ensemble of existing techniques. The primary technical novelty lies mainly in the substitution of standard SDE noise with structured alternatives. The conceptual leap from established Flow-GRPO extensions to this "Evolutionary Strategy" framework seems overclaimed.
2. The experimental evaluation is insufficient as it fails to include direct comparisons with state-of-the-art (SOTA) algorithms, TempFlow-GRPO, DiffusionNFT and BranchGRPO.
3. The papers lacks essential Reward Curves (Steps & GPU hours).
4. The method introduces an excessive number of hyperparameters.
5. There is a fundamental logical tension in the proposed sampling strategy. It is widely acknowledged in diffusion models that early denoising steps exhibit higher entropy and are more critical for structural composition. However, to maintain constant per-step wall-clock time, the authors reduce the number of candidates during these early stages due to the high cost of ODE evaluation. Would it not be more effective to prioritize sampling budget for early steps and truncate evaluations for later ones (similar to the Flow-GRPO-Fast approach)?
6. The authors claim that a KL-divergence loss is unnecessary. However, the absence of an explicit KL constraint often makes models to reward hacking. Please show Geneval when PickScore as reward.

[1] Tempflow-grpo: When timing matters for grpo in flow models

[2] BranchGRPO: Stable and Efficient GRPO with Structured Branching in Diffusion Models

[3] DiffusionNFT: Online Diffusion Reinforcement with Forward Process

**Limitations:**

yes

**Strengths And Weaknesses:**

Strength:
1. This paper proposes GS-ES, a Novel Diffusion-Native Alignment Paradigm.
2. This paper establishes a formal connection between structured noise in the action space and evolutionary search in the gradient space.

Weakness:
1. The proposed framework consists of four primary modules: Structured Noise Sampling, Multi-step ODE evaluation, Selection Mechanisms, and Gradient Aggregation. However, the overall research trajectory appears to align closely with existing extensions of the Flow-GRPO paradigm. Specifically, the use of Structured Noise Sampling and Multi-step ODE evaluation has been previously introduced and validated in TempFlow-GRPO. Regarding the Selection Mechanism, similar trajectory pruning or branching strategies have been explored in recent works such as Branch-GRPO, E-GRPO.
Consequently, the core technical novelty of this paper—replacing standard Gaussian noise in SDE transitions with structured alternatives and implementing a dynamic pruning strategy—seems to be a combination of existing techniques rather than a fundamental paradigm shift. I am concerned that the contribution is largely incremental, as it primarily refines known components.

---

> ### Author Rebuttal · Authors · 2026-03-31
>
> We sincerely thank Reviewer cVaG for acknowledging our formal connection between action-space noise and gradient-space evolution. We address your concerns below.
>
> ### 1. Conceptual Leap & Core Novelty (Weakness 1 & KeyQuestion 1)
> We acknowledge that GS-ES utilizes multi-step ODE evaluations and branching operations, components that have also been explored in recent works (e.g., TempFlow-GRPO, BranchGRPO). However, our core contribution is not merely combining these techniques to patch the standard GRPO framework.
>
> Instead, our work focuses on leveraging the unique characteristics of diffusion models—specifically, step-wise differentiability and fixed-horizon sampling—to design a fundamentally better algorithmic framework.
> We drew inspiration from the mature framework of Evolution Strategies (ES), which provided a solid foundational reference for our structural design.
> Through rigorous experiments, we have validated the effectiveness of this ES-based design.
> Furthermore, this framework is highly extensible; other heuristic algorithmic designs (such as the GS-ES-Fast approach discussed below) can be seamlessly integrated into it to achieve even better results.
>
> ### 2. Comparisons with SOTA Baselines (KeyQuestion 2)
> We agree comparisons with TempFlow-GRPO, BranchGRPO, and DiffusionNFT are essential. Due to character limits, please see our detailed theoretical and empirical comparisons in our **Response to Reviewer ADvX (Exp 4)**, demonstrating GS-ES's superior alignment performance.
>
>
> ### 3. Reward Curves: Steps (KeyQuestion 3)
> We initially omitted these curves because population-based selection yields artificially inflated training rewards, which can be visually misleading against baselines. Agreeing on their importance, we provide these plots at [Supply_Figure_2 anonymous link](https://anonymous.4open.science/r/Anonymous_figs-22153/Supply_Figure_2_reward_curve.png), displaying both the sampled rewards during training and the rewards on the validation set against optimization steps.
>
>
> ### 4. Hyperparameter Design (KeyQuestion 4)
> While our method introduces several hyperparameters, their design naturally follows the mature principles of the ES framework and straightforward computational heuristics.
> For instance, the number of candidates and selection ratios can be easily determined based on available compute budgets (e.g., constant per-step compute or early-stage prioritization).
> From this perspective, the logic behind setting these hyperparameters is highly consistent with baseline algorithms like Flow-GRPO or DanceGRPO, making them easy to adapt in practice.
>
>
> ### 5. Sampling Budget Allocation (KeyQuestion 5)
> We completely agree that early denoising steps exhibit higher entropy and dictate structural composition.
> Our initial design was based on the standard setups of Flow-GRPO and DanceGRPO. To demonstrate the effectiveness of our framework under the simplest possible conditions, we opted for a constant per-step wall-clock time (though the widely used time-shift operation inherently reflects the importance of high-noise timesteps to some extent).
>
> As you rightly suggested, prioritizing the early-step sampling budget and truncating later evaluations is far more optimal. To validate this, we implemented **GS-ES-Fast (Exp 5)**, calculating the loss *only* on the first half of the sampling window (window size=4, matching Flow-GRPO-Fast).
>
>
> | Method | PickScore ($\uparrow$) | GPU Hours (to convergence) ($\downarrow$) |
> | :--- | :--- | :--- |
> | Flow-GRPO | 23.40 | 820 |
> | GS-ES | **24.27** | 800 |
> | GS-ES-Fast | 23.89 | **390** |
>
> Integrating your suggested early-stage prioritization achieves an optimal performance-to-compute trade-off, which we will feature in the revision.
>
>
> ### 6. Reward Hacking & GenEval Evaluation (KeyQuestion 6)
> Regarding explicit KL constraints and reward hacking:
> - **Implicit Trust Region (Exp 1):** Our **Response to Reviewer HSVu (Point 2)** shows GS-ES yields a much smoother KL drift than Flow-GRPO, naturally preventing aggressive policy shifts.
> - **GenEval Evaluation (Exp 6):** We evaluated GenEval when trained *purely* on PickScore (Base: FLUX.1 Dev).
>
> | Method | Overall | Single Obj. | Two Obj. | Counting | Colors | Position | Attr. Binding |
> | :--- | :--- | :--- | :--- | :--- | :--- | :--- | :--- |
> | FLUX.1 Dev | 0.66 | 0.98 | 0.81 | 0.74 | 0.79 | 0.22 | 0.45 |
> | **GS-ES (Ours)** | **0.76** | **0.98** | **0.91** | **0.88** | **0.81** | **0.33** | **0.63** |
>
> This confirms GS-ES maintains rule-following compositionality without explicit KL constraints. For results **trained directly on GenEval**, please see our **Response to Reviewer HSVu (Point 4)**, which shows stable improvements.

---

> > ### Author Rebuttal · Reviewer_cVaG · 2026-04-04
> >
> > I appreciate the authors' efforts in providing additional experimental results. All my concerns are resolved, and I will consider raise my score.

---

### Official Review · Reviewer_ADvX · 2026-03-12

**Soundness:** 3
**Presentation:** 3
**Significance:** 3
**Originality:** 2
**Overall Recommendation:** 4
**Confidence:** 3

**Summary:**

The paper proposes **Gradient-Space Evolution Strategies (GS-ES)**, a diffusion alignment method that reinterprets action-space sampling in diffusion reinforcement learning as **gradient-space exploration**. By leveraging the duality between structured noise perturbations in the action space and parameter-space gradient directions, GS-ES proposes more stable and efficient update directions directly from diffusion noise.

The proposed method expands exploration by generating multiple candidate noise perturbations at each denoising step and selecting those that lead to better reward outcomes. This effectively reduces the high variance inherent in standard policy gradient approaches (including FlowGRPO and DanceGRPO) while also removing the need for a reference model. Empirically, the method demonstrates improved training stability, faster convergence, and better alignment performance compared to prior diffusion RL baselines.

**Compliance With Llm Reviewing Policy:**

Affirmed.

**Final Justification:**

During the rebuttal, the authors addressed my concern regarding the lack of comparisons with prior methods (e.g., BranchGRPO) and demonstrated that their approach achieves comparable performance. Therefore, I will maintain my positive score.

**Key Questions For Authors:**

Please see Weaknesses.

**Limitations:**

yes

**Strengths And Weaknesses:**

Strengths

- The paper provides an interesting analytical interpretation of diffusion RL updates by relating action-space perturbations to parameter-space gradient direction. Instead of focusing on discovering better trajectories (as in methods such as FlowGRPO), the method emphasizes finding better noise perturbations that correspond to favorable parameter update directions. This perspective is conceptually appealing and helps motivate the proposed algorithm.
- Based on this interpretation, GS-ES expands the search space by sampling multiple candidate noise perturbations at each timestep and selecting promising ones. The paper also proposes a structured sampling budget schedule to control computational cost while enabling broader exploration. This design helps balance exploration and efficiency.
- The proposed approach significantly stabilizes gradient updates during training and leads to faster convergence and improved alignment performance across the evaluated benchmarks. The empirical results suggest that increasing structured exploration during training can be beneficial for diffusion alignment.

Weaknesses
- Although the paper presents a different interpretation (gradient-space evolution), several recent works also aim to improve exploration in diffusion RL, such as **BranchGRPO [1]**. The paper would benefit from a clearer discussion of how GS-ES differs from or improves upon these approaches, both conceptually and empirically.

[1] Li, Yuming, et al. "Branchgrpo: Stable and efficient grpo with structured branching in diffusion models." *arXiv preprint arXiv:2509.06040* (2025).

---

> ### Author Rebuttal · Authors · 2026-03-31
>
> We sincerely thank Reviewer ADvX for recognizing the conceptual appeal of our Action-Gradient Duality interpretation, as well as the resulting improvements in training stability and alignment performance.
>
> We completely agree that discussing recent works provides a much clearer positioning of GS-ES within the rapidly evolving landscape of Diffusion RL.
> Below, we provide the requested conceptual and empirical comparisons.
>
> ### 1. Conceptual Comparison with Recent Diffusion-RL Baselines
> Recent works have indeed made significant strides in improving exploration and credit assignment in diffusion models, but they approach the problem from fundamentally different perspectives compared to GS-ES:
> * **BranchGRPO & TempFlowGRPO (Trajectory-level Enhancements):** Both methods build heavily upon the standard FlowGRPO/DanceGRPO pipeline, utilizing branching and multi-step ODE evaluations. Their primary focus is to fix the *credit assignment* problem within the standard diffusion GRPO framework. TempFlowGRPO focuses on precise credit assignment to intermediate actions and noise-aware adaptation of optimization intensity. BranchGRPO introduces tree-based reward aggregation and pruning to alleviate the inefficiency of sequential rollouts and the unreliability of uniformly propagating sparse terminal rewards.
> * **DiffusionNFT (Forward-Process Flow Matching):** This method takes a different route by escaping the reverse sampling process entirely. It first collects samples and then contrasts positive and negative generations to define an implicit policy improvement direction, naturally incorporating reinforcement signals into a supervised learning (SFT) objective on the *forward process*. However, by treating sampling as a black box, it sacrifices the ability to explicitly explore and intervene in the fine-grained intermediate denoising steps.
> * **GS-ES (Ours - Gradient-Space Evolution):** Unlike Branch/TempFlowGRPO, which attempt to patch the credit assignment over long trajectories, and unlike DiffusionNFT, which abandons reverse-process exploration, GS-ES introduces a paradigm shift. As discussed in our response to Reviewer HSVu (Point 1), we leverage the *step-wise differentiability and fixed-horizon sampling* of diffusion models to design a novel algorithmic pipeline inspired by **ES structural designs**. Through the Action-Gradient Duality, our structured noise perturbations in the action space are mathematically equivalent to parameter-space evolution. This allows GS-ES to perform highly efficient, stable, population-based exploration at intermediate steps. Furthermore, our framework exhibits strong extensibility, allowing new algorithmic designs to be seamlessly integrated. For instance, we can easily incorporate a truncated window update strategy (i.e., the GS-ES-Fast design, detailed in our response to Reviewer cVaG, Point 5) to further amplify computational efficiency.
>
> ### 2. Empirical Comparison (Exp 4)
> To substantiate our conceptual claims, we have supplemented our evaluation with an empirical comparison against these recent methods, as well as a baseline utilizing GRPO combined with structured noise and beam search.
>
> The table below presents the alignment performance (e.g., on PickScore) and the relative training efficiency.
>
> | Method | PickScore ($\uparrow$) | GPU Hours ($\downarrow$) |
> | :--- | :--- | :--- |
> | GRPO + Structured Noise + Beam Search | 23.65 | 800 |
> | Flow-GRPO | 23.40 | 820 |
> | DanceGRPO | 23.33 | 820 |
> | BranchGRPO | 23.57 | 620 |
> | TempFlowGRPO | 23.69 | 820 |
> | DiffusionNFT | 23.91 | 680 |
> | **GS-ES (Ours)** | **24.27** | 800 |
>
> As shown, GS-ES achieves superior alignment performance compared to recent methods, as well as the enhanced GRPO baseline, establishing a new upper bound for alignment quality.
>
> While GS-ES requires slightly more GPU hours to reach full convergence compared to the methods optimized for speed (e.g., BranchGRPO and DiffusionNFT), its computational cost is strictly on par with standard baselines like Flow-GRPO, yet it delivers vastly superior results.
> Furthermore, our framework's strong extensibility easily accommodates various efficiency-oriented designs. As a prime example detailed in our **Response to Reviewer cVaG (Point 5)**, our `GS-ES-Fast` variant achieves exactly this balance by prioritizing early-stage updates, significantly reducing GPU hours while maintaining top-tier performance.
>
> We will include this comprehensive discussion and the empirical results in the revised manuscript to better highlight the unique contributions of GS-ES.

---

> > ### Author Rebuttal · Reviewer_ADvX · 2026-04-03
> >
> > Thank the authors for their response. My concerns have been addressed. I will keep my score.

---

### Official Review · Reviewer_HSVu · 2026-03-13

**Soundness:** 3
**Presentation:** 2
**Significance:** 3
**Originality:** 2
**Overall Recommendation:** 4
**Confidence:** 3

**Summary:**

This paper presents Gradient-Space Evolution Strategies (GS-ES), a framework for aligning diffusion models with human preferences by recasting the denoising trajectory as a step-wise evolutionary optimization process. The authors argue that existing Diffusion-RL methods (e.g., Flow-GRPO, DanceGRPO) are overly LLM-centric and fail to exploit two distinctive properties of diffusion models: step-wise differentiability and fixed-horizon sampling. GS-ES operates at each denoising timestep by (1) generating a population of candidate next-states via structured noise perturbations, (2) evaluating each candidate by rolling out the remaining ODE steps to obtain a reward, (3) selecting high-fitness survivors, and (4) aggregating fitness-weighted gradients to update the model. The main theoretical claim (Proposition 4.1) shows that the REINFORCE gradient under Gaussian diffusion noise can be rewritten as an action-space perturbation projected onto the parameter space via the model's Jacobian, linking the method to Natural Evolution Strategies. A dynamic compute-balanced sampling budget and structured noise strategies (orthogonal, antithetic) are introduced. Experiments on Flux (T2I) and Wan2.1 (T2V) show substantial improvements over Flow-GRPO and DanceGRPO on PickScore, HPSv2, HPSv3, and VideoAlign, with faster convergence and lower gradient variance, all while eliminating the need for a reference model.

**Compliance With Llm Reviewing Policy:**

Affirmed.

**Final Justification:**

I still don't quite understand how this method completely eliminates the KL divergence. Despite the authors' claim that their approach forms an "implicit trust region," my understanding is that trust regions can only constrain the model's per-step updates. Under extended training, without a frozen reference model, reward hacking still seems unavoidable.

Nevertheless, I am willing to raise my score, as the authors' response addressed most of my concerns—such as the method's OOD generalization performance and its comparison against advanced forward diffusion-based methods.

**Key Questions For Authors:**

- Appendix B states that evolutionary selection forms an "implicit trust region." Can you formalize this claim, e.g., by empirically measuring the KL divergence?
- See Weaknesses.

**Limitations:**

- See Weaknesses.
- I would be happy to raise my score once my concerns have been addressed.

**Strengths And Weaknesses:**

Strengths

- The observation that diffusion models have a fundamentally different structure from autoregressive LLMs is valid. I agree that we need to design alignment algorithms that respect the properties of diffusion, rather than force-fitting LLM-centric paradigms

-  The quantitative improvements are substantial across both T2I and T2V, particularly the HPSv2 gains on video (+12.34 vs. +3.48 for DanceGRPO).

- The ablation studies systematically validate each component—dynamic budget, selection strategy, noise type.

Weaknesses

- Classical ES (including NES) is fundamentally a zeroth-order method that avoids backpropagation. GS-ES, in contrast, relies entirely on first-order backpropagation for its parameter updates. The borrowing from ES is limited to two ancillary components—structured noise and population selection—which could equally be framed as enhancements to standard policy gradient methods without invoking the ES framework. The claim of "bridging ES and RL" overpromises relative to what the method actually does.

-  Without a reference model, KL penalty, or any explicit constraint on policy drift, the method has no formal mechanism to prevent reward hacking. For example, in the bottom row of Figure 4, I found GS-ES actually hacked the reward notably more seriously compared to the baselines.

- Missing thorough evaluations on out-of-domain metrics. Without out-of-distribution benchmark evaluations, it is difficult for us to clearly assess the model's actual performance and the extent of reward hacking. At the same time, the paper does not explicitly specify the training signals used.

- Missing experiments on rule-based metrics (e.g., OCR score and Geneval score). All experiments in the paper are based on model-based metrics that measure human preference/aesthetics, and the performance of GS-ES on rule-based metrics remains unclear.

- The baseline only includes the method of directly combining GRPO with diffusion. It lacks comparisons with numerous recent algorithms specifically designed based on the properties of diffusion models.

---

> ### Author Rebuttal · Authors · 2026-03-31
>
> We thank Reviewer HSVu for recognizing our motivation, the substantial quantitative improvements, and our thorough ablations. We address your concerns below.
>
> ### 1. The Connection to Evolution Strategies (Weakness 1)
> We agree with the reviewer that GS-ES fundamentally relies on first-order backpropagation for parameter updates, unlike classical zeroth-order ES.
> Our intention is not to claim GS-ES as a purely derivative-free method, but rather to show how **ES-inspired structural designs**—specifically, population generation via structured noise and population selection—can be optimally integrated with the step-wise differentiability and fixed-horizon sampling of diffusion models.
>
> Our core theoretical contribution (Action-Gradient Duality) proves that applying structured noise in the continuous action space of Diffusion models is mathematically equivalent to parameter-space evolution.
> Furthermore, this ES framework provides a highly extensible foundation for future alignment algorithms, such as incorporating diversity-aware selection or truncated window updates (e.g., GS-ES-Fast; please refer to our discussion with Reviewer ADvX, Point 5). We will tone down the "bridging ES and RL" phrasing in the revision to avoid overpromising.
>
> ### 2. Reward Hacking & Implicit Trust Region (Weakness 2 & KeyQuestion 1)
> We acknowledge that removing the reference model increases the theoretical risk of reward hacking, which is a common phenomenon in diffusion RL.
> However, this design choice significantly reduces system complexity and memory overhead.
> In practice, standard regularization techniques (e.g., EMA, LoRA, early stopping, and hybrid sampling method in MixGRPO) effectively mitigate severe over-optimization.
>
> Regarding the "implicit trust region," we formalize this claim by empirically tracking the KL divergence during training (**Exp 1**).
> As shown in the [Supply_Figure_1 anonymous link](https://anonymous.4open.science/r/Anonymous_figs-22153/Supply_Figure_1_kl_diff.png), the KL divergence of GS-ES grows significantly slower and more smoothly than that of Flow-GRPO.
> We attribute this to the fact that GS-ES evaluates a population originating from the *same* baseline state and performs selection before gradient aggregation.
> This steady, variance-reduced parameter drift provides direct empirical evidence for the implicit trust region formed by our evolutionary selection mechanism.
>
> ### 3. Out-of-Domain Generalization (Weakness 3)
> In the original submission (Table 1), the training signals used were the corresponding in-domain reward models (e.g., trained on PickScore, evaluated on PickScore). We completely agree that Out-of-Domain (OOD) evaluations are critical for assessing true alignment and the extent of reward hacking.
>
> We have conducted **Exp 2** using PickScore as the sole training signal (In-Domain) and evaluated the model on HPSv2, HPSv3 and ClipScore (Out-of-Domain).
> As shown below, GS-ES maintains strong generalization across alternative reward models, indicating that the model learns generalized human aesthetics rather than merely exploiting the PickScore reward function.
>
> | Method (Train: PickScore) | PickScore (ID) | HPSv2 (OOD) | HPSv3 (OOD) | ClipScore (OOD) |
> | :--- | :--- | :--- | :--- | :--- |
> | Base Model | 23.03 | 29.55 | 13.91 | 37.51 |
> | DanceGRPO | 24.16 | 33.82 | 14.63 | 38.59 |
> | Flow-GRPO | 24.32 | 34.46 | 14.62 | 37.89 |
> | **GS-ES (Ours)** | **24.96** | **34.83** | **14.97** | **38.74** |
>
> ### 4. Performance on Rule-Based Metrics (Weakness 4)
> We initially omitted GenEval because the baseline (Flow-GRPO) already approaches near-saturation (0.95/1.00). To address your concern, we provide the GenEval results below (**Exp 3**).
>
> | Method | Overall | Single Obj. | Two Obj. | Counting | Colors | Position | Attr. Binding |
> | :--- | :--- | :--- | :--- | :--- | :--- | :--- | :--- |
> | SD3.5-M | 0.63 | 0.98 | 0.78 | 0.50 | 0.81 | 0.24 | 0.52 |
> | Flow-GRPO | 0.95 | **1.00** | **0.99** | 0.95 | 0.92 | **0.99** | 0.86 |
> | **GS-ES (Ours)** | **0.97** | 0.99 | **0.99** | **0.97** | **0.94** | **0.99** | **0.92** |
>
> Even in this saturated regime, GS-ES yields stable improvements. Furthermore, to explicitly address reward hacking via OOD rule-based metrics, please refer to our **Response to Reviewer cVaG (Point 6)**. Those results evaluate GenEval when the model is trained *purely on PickScore*.
>
> ### 5. Comparisons with Recent Diffusion-RL Baselines (Weakness 5)
> We agree that comparisons with recent works (e.g., BranchGRPO, TempFlowGRPO, DiffusionNFT) are essential. Due to character limits, please see our **Response to Reviewer ADvX (Exp 4)** for a comprehensive theoretical and empirical comparison demonstrating GS-ES's relative performance against these latest baselines.

---

> > ### Author Rebuttal · Reviewer_HSVu · 2026-04-04
> >
> > Thanks for the reply. My partial concerns have been addressed.
> >
> > However, I believe that demonstrating other out-of-distribution (OOD) metrics for measuring human preferences/aesthetics is insufficient when training with PickScore, as these metrics are in fact highly correlated with each other. Based on my visual inspection, the method shown in the bottom row of  Figure 4 exhibits more severe artifacts compared to the baselines. In my experience, this specific artifact pattern is precisely caused by over-optimizing PickScore.
> >
> > Based on my experience, a setting that better reflects reward hacking would be: training on OCR/GenEVAL and then evaluating on OOD metrics that measure human preferences/aesthetics. Additionally, the authors did not clarify what setting was used for the Flow-GRPO being compared—specifically, whether CFG and KL loss were used or not.
> >
> > Meanwhile, I also do not quite understand why the authors claim their method provides an implicit trust region; such a claim should at least be supported by some analysis at the update gradient or loss level.

---

> > > ### Author Response · Authors · 2026-04-06
> > >
> > > We sincerely thank you for the continued engagement and the constructive follow-up questions.
> > > Below, we address your remaining concerns in detail:
> > >
> > > **1. Reward Hacking & The GenEval -> Aesthetics Experiment**
> > > We agree that reward hacking is a common challenge in Diffusion RL.
> > > As discussed in our previous responses, we have provided a series of experiments to demonstrate how our method addresses this issue:
> > > * **Cross-Reward (Exp 2):** Point 3 in our Round 1 response.
> > > * **In-Domain GenEval Evaluation (Exp 3):** Point 4 in our Round 1 response.
> > > * **OOD Rule-Based Evaluation:** In our response to Reviewer cVaG (Point 6), confirming rule-following compositionality without explicit KL constraints.
> > > * **Qualitative Fidelity:** In our response to Reviewer L4h3 (Point 3), where we computed and provided CLIP scores for Figure 4.
> > >
> > > To further address your request for a reverse setting, which serves as a rigorous test for reward hacking, we provide the performance of the model trained purely on a rule-based signal (GenEval) evaluated on OOD aesthetic/human preference metrics (PickScore, HPSv2):
> > >
> > > | Method (Train: GenEval) | GenEval (ID) | PickScore (OOD) | HPSv2 (OOD) |
> > > | :--- | :--- | :--- | :--- |
> > > | Base Model (SD3.5-M) | 0.63 | 22.80 | 29.81 |
> > > | Flow-GRPO | 0.95 | 22.57 | 27.44 |
> > > | **GS-ES (Ours)** | **0.97** | 22.64 | 28.99 |
> > >
> > > As observed in the table, achieving substantial gains in rule-following capabilities (GenEval) inherently introduces a trade-off, leading to a natural degradation in purely aesthetic metrics compared to the base model.
> > > However, GS-ES demonstrates superior stability in this trade-off: it not only achieves higher rule-following accuracy but also preserves aesthetic quality better than Flow-GRPO.
> > > This further confirms that our method effectively optimizes the target reward while maintaining greater robustness against severe over-optimization.
> > >
> > > Furthermore, as noted in our initial rebuttal, in practice, standard regularization techniques (e.g., EMA, LoRA, early stopping, and hybrid sampling methods as in MixGRPO) are also commonly and effectively employed to mitigate severe over-optimization in diffusion RL.
> > >
> > > **2. Baseline Experimental Settings (Flow-GRPO)**
> > > We apologize for omitting these details in the initial manuscript.
> > > To ensure a fair and rigorous comparison, we strictly followed the official codebase and community-standard settings for the Flow-GRPO baseline. We will explicitly include these hyperparameter settings in the revised manuscript.
> > >
> > > Specifically, in our comparisons, Flow-GRPO **did** utilize both Classifier-Free Guidance (CFG) and a KL divergence penalty. The specific hyperparameters used for Flow-GRPO were:
> > > * **CFG Scale:** 3.5
> > > * **KL Loss Coefficient (beta):** 0.04
> > >
> > > **3. The "Implicit Trust Region" at the Gradient/Loss Level**
> > > We argue that in Diffusion RL, the primary purpose of a KL divergence penalty is to prevent the model from taking extreme gradient steps that destroy the original distribution (collapse).
> > > GS-ES achieves this stability without an explicit KL loss through its variance-reduced gradient estimation mechanism, which is supported by both empirical evidence and theoretical analysis:
> > >
> > > * **Empirical Evidence:**
> > >     * **Gradient Stability (Figure 3, Lines 372-384):** In our original manuscript, we empirically verified this by plotting the sliding standard deviation of the gradient norms. As observed, GS-ES exhibits consistently lower volatility compared to DanceGRPO. Given that both methods maintain comparable gradient magnitudes, this lower deviation serves as a direct indicator of a more stable gradient estimator.
> > >     * **KL Loss Trajectory (Exp 1 from Rebuttal Round 1):** This gradient-level stability translates directly to training dynamics. As shown in Exp 1, the KL divergence during GS-ES training increases smoothly and steadily without sudden spikes or policy collapse, proving that the model updates stably without wildly deviating from the base policy.
> > >
> > > * **Theoretical Mechanism (Sec 4.3):**
> > >     * **Standard Diffusion RL** (e.g., Flow-GRPO) averages gradients ($\nabla_\theta \log \pi_\theta(x_{t-1}|x_t)$) across entirely independent rollouts with different intermediate states ($x_t$). This macroscopic averaging naturally introduces high variance.
> > >     * **GS-ES (Ours)**, as proven in **Proposition 4.1**, performs an evolutionary update directly in the gradient space. At each timestep, we average gradients over a population of structured noise perturbations originating from the *exact same state $x_t$*.
> > >     * **The Result:** This localized, action-space sampling is projected via the model's Jacobian into a naturally bounded, variance-reduced parameter update. By anchoring exploration at a shared state, GS-ES inherently restricts the variance of the update steps. This forms an "implicit trust region" that ensures training stability without needing an artificial KL penalty.

---

### Decision · Program_Chairs · 2026-04-30

**Decision:**

Reject

**Comment:**

This paper proposes an alignment framework for diffusion model that reinterprets action-space sampling as gradient-space exploration and recasts the denoising trajectory as a step-wise evolutionary optimization process. The formulation reduces the high variance inherent in standard policy gradients while eliminating the computational burden of maintaining reference models. Empirical benefit was demonstrated. Although the method is heuristic, reviewers and AC found the idea interesting. Meanwhile, there are concerns about overclaim, empirical evaluation, reward hacking, and not-up-to-date survey and comparison with the literature. Unfortunately, these concerns have not been fully resolved by the end of the rebuttal discussion. Recognizing the potential of the idea, I encourage the authors to take the discussions into consideration and re-submit a revised version.